# The Evolving Role of Neutrophils and Neutrophil Extracellular Traps (NETs) in Obesity and Related Diseases: Recent Insights and Advances

**DOI:** 10.3390/ijms252413633

**Published:** 2024-12-20

**Authors:** Serena Altamura, Francesca Lombardi, Paola Palumbo, Benedetta Cinque, Claudio Ferri, Rita Del Pinto, Davide Pietropaoli

**Affiliations:** 1Department of Life, Health & Environmental Sciences, University of L’Aquila, 67100 L’Aquila, Italy; serena.altamura@graduate.univaq.it (S.A.); francesca.lombardi@univaq.it (F.L.); paola.palumbo@univaq.it (P.P.); benedetta.cinque@univaq.it (B.C.); claudio.ferri@univaq.it (C.F.); rita.delpinto@univaq.it (R.D.P.); 2Prevention and Translational Research—Dental Clinic, Center of Oral Diseases, 67100 L’Aquila, Italy; 3Unit of Internal Medicine and Nephrology, San Salvatore Hospital, Center for Hypertension and Cardiovascular Prevention, 67100 L’Aquila, Italy

**Keywords:** obesity, obesity-related diseases, neutrophils, neutrophil extracellular traps, NETosis, inflammation, gut microbiome, dysbiosis, short-chain fatty acids

## Abstract

Obesity is a chronic, multifactorial disease characterized by persistent low-grade tissue and systemic inflammation. Fat accumulation in adipose tissue (AT) leads to stress and dysfunctional adipocytes, along with the infiltration of immune cells, which initiates and sustains inflammation. Neutrophils are the first immune cells to infiltrate AT during high-fat diet (HFD)-induced obesity. Emerging evidence suggests that the formation and release of neutrophil extracellular traps (NETs) play a significant role in the progression of obesity and related diseases. Additionally, obesity is associated with an imbalance in gut microbiota and increased intestinal barrier permeability, resulting in the translocation of live bacteria, bacterial deoxyribonucleic acid (DNA), lipopolysaccharides (LPS), and pro-inflammatory cytokines into the bloodstream and AT, thereby contributing to metabolic inflammation. Recent research has also shown that short-chain fatty acids (SCFAs), produced by gut microbiota, can influence various functions of neutrophils, including their activation, migration, and the generation of inflammatory mediators. This review comprehensively summarizes recent advancements in understanding the role of neutrophils and NET formation in the pathophysiology of obesity and related disorders while also focusing on updated potential therapeutic approaches targeting NETs based on studies conducted in humans and animal models.

## 1. Introduction

Obesity is a complex metabolic disease influenced by various factors, including genetics, socioeconomic status, and environmental conditions. These elements affect food consumption, nutrient absorption, thermogenesis, and fat storage in adipose tissue (AT) [1,2,3]. Obesity is linked to conditions such as hyperglycemia, dyslipidemia, and hypertension, making it a major risk factor for various conditions. These include insulin resistance (IR), type 2 diabetes (T2D), impaired kidney function, poor wound healing, non-alcoholic steatohepatitis (NASH), cardiovascular disease, Alzheimer’s disease, stroke, sleep apnea, cancer, and others [4]. The metabolism of AT plays a crucial role in the negative effects of increased body fat [5,6].

Chronic low-grade inflammation is a hallmark of metabolic disorders like obesity [7]. Excessive caloric intake and subsequent weight gain lead to an increase in the number or size of adipocytes. This results in changes to the phenotype of AT, particularly in white AT. This shift is characterized by the development of inflamed adipocytes with altered functions and the recruitment of immune cells that release pro-inflammatory cytokines [8]. In lean mice, AT inflammation is suppressed by a coordinated interaction between immunosuppressive CD4^+^ regulatory T cells (Tregs), anti-inflammatory M2 macrophages, type 2 innate lymphoid (ILC2) cells, and eosinophils. In contrast, obesity causes a dramatic increase in pro-inflammatory CD4^+^ type 1 T helper (Th1) cells, cytotoxic CD8^+^ T cells, and M1 macrophages, while the number of Tregs decreases, resulting in chronic low-grade inflammation [8].

Although human data is still limited, evidence suggests a similar inflammatory response involving immune cells in the AT of obese individuals. In cases of obesity, peripheral neutrophils are often the first immune cells to infiltrate AT in high-fat diet (HFD) conditions [9]. An HFD can lead to a twenty-fold increase in neutrophil numbers within three days, compared to seven days for macrophages [9]. Additionally, obesity is known to alter the gut microbiome and increase gut permeability [10], facilitating the translocation of endotoxins, such as lipopolysaccharides (LPS), which drive inflammation in various tissues, including AT [11]. Pathogenic bacteria from the gastrointestinal tract may also translocate into obese AT, contributing to increased neutrophil activity [12]. Infiltrating neutrophils utilize various mechanisms, including phagocytosis, reactive oxygen species (ROS) production, degranulation, and the release of neutrophil extracellular traps (NETs). While NETs are essential for the antimicrobial response of neutrophils [13], excessive release of these traps can lead to vessel occlusion, tissue damage, and an exaggerated immune response due to ROS production and the release of cytotoxic granule proteins [13,14]. Emerging evidence suggests that elevated NET levels may contribute to the progression of obesity and its associated comorbidities. Studies indicate that deleting or inhibiting NETs may be a safe approach for these patients [15,16,17]. Therefore, targeting NETs is crucial for managing inflammation related to obesity. This review emphasizes the role of neutrophils in inflammation associated with obesity, highlighting the significance of neutrophil infiltration in the immune dysfunction resulting from obesity. Furthermore, we discuss recent findings regarding the mechanisms behind the formation of NETs and their implications for obesity and related diseases.

## 2. Review Criteria

This review is based on original articles and reviews published over the last two decades, which were retrieved through PubMed using specific search terms or combinations thereof: obesity, inflammation, neutrophils, NETs, NE, MPO, endothelial dysfunction, type 2 diabetes (T2D), LPS, gut microbiota, and various outcomes. Only papers published in English were included in the review. Additionally, further relevant papers were identified from the reference lists of the retrieved articles.

## 3. Obesity-Related Peripheral Neutrophilia

Polymorphonuclear neutrophils are the most abundant type of white blood cells in human blood and play a critical role as primary effector cells in the human immune system. They protect the body against pathogens and inflammatory stimuli [18,19,20]. When an infection or inflammation occurs, neutrophils migrate from the bloodstream to the affected sites, acting as phagocytic cells. They release proteolytic enzymes through a process called degranulation, produce ROS through oxidative bursts, and form NETs that possess antimicrobial properties [18]. In addition to their role in fighting infections, neutrophils are essential participants in acute inflammatory reactions, as they are the first leukocytes to be recruited to sites of inflammation. They produce significant amounts of cytokines and chemokines, such as tumor necrosis factor-alfa (TNF-α), interleukin-1beta (IL-1β), interleukin-8 (IL-8), and monocyte chemoattractant protein-1 (MCP-1), all of which help regulate the overall immune response [21].

Moreover, neutrophils play a crucial role in regulating adaptive immunity by recruiting and activating T lymphocytes at inflammation sites [22]. They also produce various cytokines that influence lymphocytes and can act as antigen-presenting cells (APCs) [22]. Research focusing on obesity has shown that increased adiposity is associated with increased levels of leukocytes in circulation. Specifically, neutrophil counts have been found to positively correlate with body mass index (BMI), waist circumference, and serum C-reactive protein (CRP) levels [23,24,25,26,27]. In individuals with obesity, neutrophils exhibit an activated phenotype, indicated by increased plasma concentrations of myeloperoxidase (MPO) and neutrophil elastase (NE) compared to healthy controls [28,29,30,31]. This heightened activity of neutrophils is linked to a greater release of pro-inflammatory cytokines and cardiovascular risk biomarkers [29]. For instance, in humans, plasma MPO levels have been found to positively correlate with endothelial dysfunction [32] and cardiovascular disease risk in individuals with T2D [33]. Collectively, these findings underscore the significance of neutrophils and their activation in the development of obesity-related diseases, highlighting a critical event in the inflammatory response to HFDs.

## 4. Neutrophils: The First Immune Cells Infiltrating Adipose Tissue (AT) During Obesity

While macrophages are the most studied and undeniably crucial immune cells involved in AT inflammation, neutrophils are emerging as critical early players in the pathogenesis of obesity. Under normal, homeostatic conditions, only a small number of neutrophils are present in AT. However, during inflammation induced by experimental obesity, the number of neutrophils infiltrating AT increases rapidly [9]. Peripheral neutrophils are recruited to AT through various chemotactic factors produced in the tissue [34]. In the context of obesity, inflamed adipocytes produce elevated levels of IL-8, which acts as a potent chemoattractant for neutrophil infiltration into AT (Figure 1).

Additionally, neutrophils can attract more circulating neutrophils by releasing C–X–C motif chemokine ligand 2 (CXCL2), another significant neutrophil chemoattractant [35,36]. Once in the AT, neutrophils interact with adipocytes by binding integrin αMβ2 (also known as Mac-1), which is present on the neutrophils, to intercellular adhesion molecule 1 (ICAM-1) on the adipocytes [9]. Recent studies have reported a bidirectional crosstalk between adipocytes and neutrophils, which contributes to AT inflammation through the production of various pro-inflammatory substances, such as leptin, IL-1β, TNF-α, ROS, matrix metalloproteinase-8 (MMP-8), often involving infiltrating macrophages [37,38]. In fact, neutrophils, when activated, release inflammatory factors that recruit macrophages and other immune cells, including B cells, T cells, and natural killer (NK) cells. This process perpetuates the inflammatory state as these cells produce cytokines and chemokines that can spread to other parts of the body, creating a systemic inflammatory condition. In obese AT, macrophages increase significantly due to enhanced recruitment, retention, and proliferation. Recruitment is mainly mediated by the myeloid C–C motif chemokine receptor-2 (CCR2), which is the receptor for MCP-1. Macrophage retention involves direct contact with adipocytes, facilitated by the adhesion of integrin α4β1 on macrophages to VCAM-1 on adipocytes. Macrophages can also proliferate in response to Th2 cytokines, such as IL-4, IL-13, and granulocyte/monocyte colony-stimulating factor (GM-CSF). Additionally, neutrophils interact directly with adipocytes through the binding of integrin αMβ2 to ICAM-1 on the adipocytes [9]. This interaction leads to the production of IL-1β and TNF-α, which are important activators of macrophages [39,40,41]. Neutrophils also produce NE, which directly activates macrophages, and granule protein cathelicidin (LL-37), which can stimulate the release of additional proinflammatory cytokines from macrophages. Activated neutrophils can further recruit monocytes through the release of azurocidin, LL-37, cathepsin G, proteinase 3 (PR3), and human neutrophil peptides 1–3 (HNP1–3) [42,43]. Furthermore, lactoferrin, azurocidin, and HNP1–3 can induce the polarization of macrophages toward the M1 proinflammatory phenotype [44,45,46]. Therefore, neutrophil proteins play a significant role in intensifying inflammation by promoting macrophage activation and the subsequent release of proinflammatory cytokines.

In animal models of obesity-induced inflammation, the number of neutrophils in peripheral circulation increases, allowing them to infiltrate the blood vessel endothelium and AT [9,47]. Notably, neutrophil infiltration into AT can occur as early as three days after HFD feeding and can be sustained for up to 90 days, indicating that neutrophils play a crucial role in the early stages of obesity [48]. This early inflammatory response dominated by neutrophils has been shown to contribute in the pathogenesis of metabolic alterations that accompany obesity [9,15,27,48,49,50,51,52,53,54,55,56,57].

A study by Mansuy-Aubert et al. utilized a proteomic approach and found that serum levels of alpha-1 antitrypsin (A1AT), an endogenous inhibitor of NE, were significantly decreased in both the serum and liver of obese mice [15]. In contrast, the researchers observed that NE activity was notably elevated in the serum of both obese and HFD-fed mice. This suggests that obesity is linked to a substantial increase in the ratio of NE protease activity to its natural inhibitor, A1AT. Notably, genetic deletion of NE and overexpressing of human A1AT significantly reduced AT inflammation, IR, body weight gain, and liver steatosis in mice fed an HFD [15,48]. Furthermore, Kawanishi et al. demonstrated that exercise training reduced both neutrophil infiltration and NE expression in the AT of HFD-induced obese mice [51], leading to a decrease in macrophage content and inflammatory cytokine expression in the AT.

In a recent study conducted by Shantaram et al., researchers gavaged microbiome-depleted mice with stool samples from individuals with and without obesity during high-fat or normal diet administration. The results showed that only the mice on an HFD that received stool from subjects with obesity exhibited an enrichment of VAT neutrophils, suggesting donor microbiome and recipient HFD determine VAT neutrophilia [12]. Interestingly, transcriptomic analyses revealed that VAT neutrophils are functionally distinct from circulating neutrophils [12]. VAT neutrophils also showed a significant increase in the production of extracellular matrix components and other proteins that influence their surrounding microenvironment [12]. These components included growth factors, fibrosin, angiopoietin-2, amphiregulin, various types of collagen, MMPs, and tissue inhibitors of MMPs. Furthermore, the transcriptomic analysis indicated that VAT neutrophils exhibited higher expression of genes associated with ROS production and degranulation, suggesting an enhanced pro-inflammatory function [12]. This distinct profile implies that neutrophils in VAT adapt to a lipid-rich environment while expressing genes that promote inflammation.

Table 1 summarizes the main preclinical and clinical studies investigating the role of activated neutrophils in obesity and related diseases.

## 5. Neutrophil Extracellular Traps (NETs)

One of the key functions of neutrophils is the formation of NETs [59,60,61]. NETs are composed of proteins from azurophilic (primary) granules, which include NE, cathepsin G, and MPO. They also contain proteins from secondary and tertiary granules, such as lactoferrin and gelatinase [62,63]. However, the primary component of NETs is nuclear DNA [64]. This web-like structure enables NETs to prevent the spread of pathogens throughout the body, functioning effectively as traps that capture, localize, and eliminate these invaders [64]. Originally, the release of a NET was linked to neutrophil cell death, a process termed “NETosis”. However, recent expert reviews suggest that “NETosis” may not fully encompass all types of NET release. As a result, they recommend using the term “NET formation”, especially when neutrophil death is not observed [65].

NET formation can occur in two ways: with neutrophil cell death, known as “suicidal NETosis”, or while preserving neutrophil functions, known as “vital NETosis”. There is a broad consensus that the formation of NETs in response to both microbial and sterile agents is a natural phenomenon occurring in vivo. Nonetheless, only a limited number of studies have addressed the direct effects of specific stimuli on NET induction in vivo. Despite these limitations, various infectious and sterile stimuli have been identified as triggers for NET formation. These stimuli include a range of pathogens, cytokines (such as IL-8, IL-1β, and TNF-α), chemokines, immune complexes, and interactions between neutrophils and activated platelets or endothelial cells (ECs) [66]. Additionally, several non-physiological agonists, including bacterial products like LPS and phorbol esters like phorbol myristate acetate (PMA), are commonly used to induce NETs ex vivo [67].

## 6. Pathways of NET Formation

NET formation occurs through two main pathways: the nicotinamide adenine dinucleotide phosphate (NADPH) oxidase 2 (Nox2)-dependent pathway and the Nox2-independent pathway [68]. One of the best-understood mechanisms leading to NET release is the lytic pathway. This process requires Nox2 activity and consists of several steps: ceasing actin rearrangement and polymerization, disassembling the nuclear envelope, and decondensing nuclear chromatin. These steps lead to the mixing of chromatin with the cytoplasm, ultimately resulting in the release of DNA into the extracellular space. Stimulation of neutrophils by PMA, LPS, and bacteria activates Nox2 via the protein kinase C (PKC) and c-Raf-MEK-Akt-ERK signaling pathway [69]. Activation of Nox2 generates ROS and increases calcium influx, which in turn activates protein arginine deiminase 4 (PAD4) [70]. PAD4 hypercitrullinates histones H3, H2A, and H4, resulting in reduced positive charge on the histones and subsequent chromatin decondensation [71]. Simultaneously, the formation of ROS promotes the translocation of MPO and NE, two key enzymes stored in the azurophilic granules of naïve neutrophils, into the nucleus, which aids in the disruption of chromatin packaging [72]. MPO converts hydrogen peroxide into hypochlorous acid, which activates NE. This activation results in the degradation of the cytoskeleton and the dismantling of the nuclear membrane, facilitating the expulsion of NETs [73]. During PMA-induced NET release, NE cleaves gasdermin D to its active form [74], which forms pores in the plasma and granule membranes, enhancing the release of NE and other granule contents [75]. The lytic pathway lasts several hours (1–4 h) and when it leads to neutrophil death, it is referred to as “suicidal NETosis” [76].

In contrast, the non-lytic pathway of NET formation shows that histone citrullination and chromatin decondensation can occur independently of Nox2 through the activation of PAD4 [70,77,78]. This pathway can be initiated by calcium ionophores, such as ionomycin, which increase intracellular calcium levels and activate PAD4 to facilitate histone H3 citrullination [79]. As a result, the electrostatic bond between histones and DNA weakens, leading to chromatin decondensation [80,81]. The modified chromatin is then packaged into vesicles that merge with the plasma membrane, allowing the nuclear DNA to be released outside the cell through vesicular transport without damaging the plasma membrane. This mechanism allows neutrophils to maintain their viability and normal functions, including chemotaxis and phagocytosis [82]. This pathway, referred to as “vital NETosis”, allows neutrophils to release NETs within 5 to 60 min, independent of ROS and the Raf/MEK/ERK pathway [60,68].

## 7. NETs in Obesity and Obesity-Related Diseases

Obesity-induced low-grade chronic inflammation plays a crucial role in activating neutrophils. Therefore, it is important to explore the potential link between obesity, related diseases (i.e., T2D and cancer), inflammation, and NETs. Several experimental and human studies have shown a strong association between obesity-related inflammation and elevated levels of NETs. In a diet-induced obesity (DIO) mouse model, researchers observed increased plasma concentrations of MCP-1/CCL2 [16]. MCP-1 is a potent chemotactic factor for monocytes and is known to contribute to diseases associated with endothelial dysfunction, such as atherosclerosis [83]. The authors suggest that since MCP-1 expression may be connected to cardiovascular disease progression associated with obesity, their findings provide further support for the hypothesis that NET formation is involved in the inflammatory processes linked to DIO. Additionally, circulating concentrations of MCP-1 are found to be higher in obese patients [84,85,86]. Moreover, cathelicidin-related antimicrobial peptide (CRAMP), a marker of NET formation [50], was significantly increased in the mesenteric arterial walls of DIO mice compared to control mice [16]. Notably, inhibiting or degrading NETs significantly reduced MCP-1 levels in DIO mice, suggesting that NET formation is a key driver of the inflammatory processes associated with obesity [16]. Indeed, prevention of NET formation with Cl-amidine or dissolution of NETs with DNase restored endothelium-dependent vasodilation to the mesenteric arteries of DIO mice. A study conducted by Van Bruggen et al. investigated the effects of PAD4 activity and NET release on hematological and cardiac physiology in the context of obesity-induced chronic inflammation [87]. The researchers utilized an animal model comprising wild-type (WT) mice and neutrophil-specific PAD4 deficient (Ne-PAD4^−/−^) mice. Both groups were fed an HFD for 10 weeks, alongside a control diet group. The findings revealed that neutrophils isolated from the HFD-fed mice exhibited increased levels of NET formation compared to those from mice on a control diet. Notably, Ne-PAD4^−/−^ mice on the HFD gained less weight than their WT counterparts, who steadily gained weight throughout the 10-week experimental period. Furthermore, WT mice showed a decrease in diastolic function when compared to baseline values, while Ne-PAD4^−/−^ mice maintained their baseline values. The authors highlighted the significance of systemic NET release during metabolic stress and its contribution to cardiac deterioration, linked to changes in the innate immune system triggered by obesity [87]. Recent studies have revealed that NET levels are significantly higher in the blood and AT of obese patients compared to those with a normal weight (eutrophic controls). Bioinformatics and proteomics analyses identified IL-8, heat shock protein 90 (HSP90), and the E1 heat shock protein family (HSPE1) as being associated with obesity, inflammation, and the release of NETs [17]. These findings suggest that the elevated NET levels in the serum of obese patients may influence inflammatory markers. As a result, inhibiting NETs could represent a potential therapeutic approach for treating obesity-related comorbidities [17].

In a study conducted by D’Abbondanza et al., researchers analyzed levels of NET byproducts in morbidly obese individuals undergoing bariatric surgery compared to healthy controls [88]. The findings revealed that the accumulation of DNA fragments associated with MPO was significantly higher in patients prior to the intervention compared to the controls (*p* < 0.001). This accumulation correlated with various factors, including body weight, BMI, waist and hip circumferences, glyco-metabolic variables, and systolic blood pressure. However, the trend of NET release following surgery was heterogeneous. Some patients showed significant improvements in NET regulation after bariatric surgery, while others did not exhibit changes in NET accumulation. Among the latter group were individuals with a higher incidence of cardiovascular events, thus supporting the possible role of non-fat associated stimuli in neutrophil activation and NET formation [88].

Several studies conducted in both humans and mice have emphasized the role of NETs in HFD-associated NASH. In a mouse model fed an HFD, abnormal neutrophil infiltration into liver tissue and subsequent NET formation were identified as early events in the development of NASH, potentially contributing to inflammation and liver damage [89,90]. Notably, the depletion of NETs using DNase I significantly reduces hepatic inflammation and slows the progression of NASH resulting from HFD-induced liver injury [89,90]. In particular, in their study, Wu et al. [90] utilized a diet-induced NASH mouse model to investigate the immune cell profile at various time points during NASH. They observed neutrophil infiltration after three weeks, along with the formation of NETs in the liver, which triggered immune responses in macrophages rather than causing injury to hepatocytes. Using a metabolomic approach, the researchers found that the linoleic acid metabolism pathway was altered in NASH mice, a finding that was further validated by reviewing existing clinical data from NASH patients. Additionally, they reported that linoleic acid and gamma-linolenic acid induced in vitro significant NETosis (*p* < 0.001 compared to control) by triggering an oxidative burst. Furthermore, they found that silybin, a hepatoprotective agent, could significantly inhibit NETosis both in vitro and in NASH mice.

In the study by van der Windt et al., elevated levels of a NET marker (MPO-DNA complexes) were observed in the serum of patients with NASH [91]. The authors also reported that neutrophil infiltration and NET formation in murine NASH models contribute to the progression of hepatocellular carcinoma. An influx of neutrophils in the livers of NASH mice was observed starting at 5 weeks, leading to an approximately two-fold increase in their numbers. Western blot analysis showed the presence of citrullinated histone-3, an indicator of NET formation, in these mouse livers. The authors noted that while neutrophil counts returned to baseline levels by 12 weeks, NET formation remained detectable throughout the 20-week duration of the experiment. Importantly, this NET formation was followed by an influx of macrophages derived from monocytes, which were recruited to the liver as part of the inflammatory response. These macrophages have been identified as the predominant effector immune cells in NASH and are a significant source of inflammatory cytokines, which contribute to the amplification of inflammation [92,93]. Importantly, inhibiting NET formation—either through DNase treatment or by utilizing PAD4 knockout mice—did not prevent the development of fatty liver, but it altered the pattern of liver inflammation and resulted in reduced tumor growth [91]. In particular, the DNase treatment resulted in a significant 45% decrease in the NASH score (*p* < 0.05), indicating a meaningful improvement in hepatic inflammation and effectively reducing inflammatory processes associated with NASH. Another study showed that increased plasma levels of NETs in NASH patients exhibited cytotoxic effects on ECs, causing them to adopt pro-coagulant and pro-inflammatory characteristics [94]. The authors observed a significant increase in circulating NET biomarkers in the plasma of NASH patients compared to healthy controls. These biomarkers included cell free-DNA (cf-DNA), MPO-DNA complexes, NE-DNA complexes, and citH3-DNA complexes, as measured using ELISA. The increase ranged from approximately two to three times, with a *p*-value < 0.001. Additionally, the levels of plasma IL-6 and TNF-α were significantly higher in patients with NASH compared to the control group. The study found a positive correlation between the levels of inflammatory factors and the markers of NETs. Furthermore, the authors observed that the impaired vascular endothelium transitioned to a pro-inflammatory phenotype, indicated by the high expression of ICAM-1 and VCAM-1. These findings support the hypothesis that the chronic inflammatory environment associated with NASH promotes the formation of NETs, which contribute to a hypercoagulant state. However, treatment with DNase I successfully reversed these effects [94]. Recent immunohistochemical studies have shown that the density of NETs is lower in colorectal cancer tissues from diabetic patients receiving metformin. These findings suggest that metformin treatment may enhance outcomes for patients with both colorectal cancer and T2D by influencing the formation of NETs [95]. In the study of Carestia et al., patients diagnosed with T2D exhibited higher levels of NET generation, as well as increased levels of NE, mono- and oligonucleotides, and cf-DNA compared to healthy individuals, and all neutrophil responses were restored to normal after 12 months of metformin treatment [96]. The authors suggest that NETs could represent a novel biomarker for T2DM and that increased in vivo NET formation appears not to be the consequence of impaired glycemic control and is not associated with thrombotic events. The different results reported in the study by Bryk et al. [97] following treatment with metformin have been attributed by the authors to factors such as the patients’ age, the timing of their diagnosis, and their history of thrombotic events. Nevertheless, the authors conclude that their findings suggest that in T2DM patients, markers of NETosis in circulating plasma, including H3Cit and cf-DNA, are associated with glycemic control, markers of systemic low-grade inflammation, and previous myocardial infarction. Increased NETosis, as detectable in circulating blood, is linked to a prothrombotic state, particularly characterized by hypofibrinolysis in T2D patients. This study indicates that NETosis may contribute to the thrombotic and cardiovascular risks associated with the disease, highlighting the need for further investigation into the role of NET generation in the natural progression of diabetes and its complications.

Menegazzo et al. identified a positive correlation between glucose control markers, such as hemoglobin A1c, and circulating NET markers, including mononucleosomes and oligonucleotides [98]. Hyperglycemia was linked to increased NET formation, suggesting a connection between diabetes and NETosis. In vitro results confirmed that high concentrations of glucose induced an increased NET release by white blood cells isolated from healthy donors [98]. Notably, treatment with metformin in patients with T2D significantly reduced concentrations of NET components [99]. On the other hand, a subsequent study by Menegazzo et al. reported that metformin treatment reduced levels of serum NE, proteinase-3, citrullinated histone, and double strand DNA (dsDNA) [99]. The study demonstrated that metformin decreased NETosis in neutrophils when exposed to classical NET-inducing agents, and importantly, this effect was independent of metformin’s anti-hyperglycemic properties [99]. Additionally, the potential clinical significance of metformin’s inhibition of neutrophils and NET formation has been explored in animal models and diabetic patients [100,101]. In summary, metformin has been shown to lower the risk of adverse cardiovascular outcomes in diabetic patients through mechanisms unrelated to glycemic control and potentially involving the inhibition of NETosis. A recent study reported that PMA and recombinant High Mobility Group Box-1 (HMGB1) effectively induced the formation of NETs from human neutrophils. This was evidenced by flow cytometric detection of citrullinated histone H3 immunopositivity. While metformin had a limited effect on spontaneous NET formation, it significantly reduced it when neutrophils were stimulated with either PMA or HMGB1 [102]. PAD4 expression also significantly increases in individuals with T2D, thereby contributing to impaired wound healing in both mice and humans [103,104]. In particular, Wong et al. found that Western blotting showed a four-fold increase in PAD4 protein expression in the neutrophils of individuals with diabetes compared to healthy controls [103]. A proteomic analysis conducted by Fadini et al. showed that NET components were more abundant in the blood of patients with nonhealing diabetic foot ulcers [104]. Elevated levels of circulating NE and PR3 were associated with infections, and serum NE levels were predictive of delayed healing [104]. Indeed, NE, a key marker of NETs, was found to be 59% higher in worsening wounds compared to those that were stable or healed (*p* < 0.05). This indicates that local NETosis is linked to impaired wound healing. In diabetic mice, increased PAD4 activity in the skin, along with evidence of histone citrullination and intravital microscopy, indicated that NETosis was occurring in the beds of excisional wounds [104]. Remarkably, inhibiting NET formation through PAD4 knockout or disrupting NETs with DNase I or Cl-amidine accelerated wound healing in diabetic mice [103,104]. Collectively, these findings suggest that NETosis may hinder the healing process of diabetic foot ulcers. Recent research also demonstrated that extracellular vesicles derived from mesenchymal stem cells can transfer functional mitochondria to neutrophils in wound tissue [105]. This transfer triggers mitochondrial fusion and restores mitochondrial function, ultimately leading to a reduction in NET formation [105]. These insights reveal a novel NET-mediated pathway involved in wound healing in diabetes and suggest that inhibiting NET formation could serve as an effective therapeutic strategy for enhancing wound healing.

Increased deposition of NETs has been reported in the glomeruli of patients with diabetic nephropathy, as well as in diabetic mice [106]. In particular, a significant increase in serum MPO-DNA complexes (*p* < 0.0001) was observed in patients with type II diabetes who also have diabetic kidney disease when compared to control subjects (patients with type II diabetes without diabetic kidney disease). The treatment of these mice with DNase I reduced their susceptibility to glomerulopathy and mitigated damage to the glomerular ECs (GECS) by lowering NET release [106].

A recent study by Wang et al. found that increased neutrophil infiltration and the formation of NETs in the VAT of obese mice contribute to the development of pancreatic cancer [107]. The results showed that obesity significantly increased the infiltration of neutrophils around visceral adipocytes. This was accompanied by a decrease in CD8+ T cells and an increase in regulatory T cells within the pancreas. However, there was no significant difference in the infiltration of tumor-associated macrophages, although a slight increase in polarization toward the M2 pro-tumor phenotype was observed in obese mice. Notably, NETs were significantly upregulated in obese mice, as indicated by higher levels of citrullinated histone H3 and MPO. Additionally, the mRNA levels of enzymes associated with NETs, such as MPO, MMP-9, NE, cathepsin G, and lactoferrin, were also elevated in these mice. Importantly, PD-L1 (Programmed Death-Ligand 1), an immune checkpoint involved in creating a tumor immunosuppressive microenvironment, was expressed in most of the infiltrating neutrophils. Metformin treatment significantly reduced the levels of NETs in the HFD group. Additionally, both metformin and DNase I were shown to significantly inhibit this process and decrease the formation of precancerous lesions [107]. Furthermore, researchers have suggested that neutrophil infiltration induced by adipocytes may contribute to desmoplasia and chemotherapy resistance in pancreatic cancer, particularly in obese mouse models [108]. These findings provide valuable insights into how neutrophils and NET formation can promote pancreatic cancer and suggest potential strategies for addressing pancreatic cancer related to obesity.

Table 2 summarizes the main preclinical and clinical studies investigating the role of NETs in obesity and related diseases. In Table 3 are reported studies and therapeutic approaches used to target NETs.

Overall, these reports highlight the significant role of NETs in obesity and obesity-related conditions, emphasizing the complex nature of obesity itself. A deeper understanding of these dynamics is essential for developing effective management strategies for obesity and its related health risks.

## 8. Obesity, Gut Dysbiosis, and NETs

Gut dysbiosis refers to a change in the gut microbiome composition that is typically characterized by reduced microbial diversity and altered microbial functions and metabolism. This condition is associated with the development of obesity and related metabolic disorders, including T2D, MetS, and cardiovascular diseases, and chronic kidney disease [109,110,111,112,113,114]. Dysbiosis is especially important in promoting chronic low-grade inflammation, also known as meta-inflammation, which is considered a significant contributor to the onset of obesity and its associated diseases [111,115,116,117].

The gut microbiota play an increasingly recognized role in maintaining energy balance and activating the host’s immune response through various molecular interactions [118]. Specific microorganisms within the gut microbiota actively regulate metabolic processes such as glucose metabolism, lipid control, and insulin sensitivity by producing metabolites like SCFAs, bile acids, and other bioactive substances [119]. Obesity, like several other human disorders, is associated with gastrointestinal dysbiosis, where the ratios, abundance, and composition of microbes are altered, subsequently affecting SCFA generation. While no definitive microbiome signature for diagnosing obesity has been identified so far, common profiles include a reduction in the abundance of species capable of producing SCFAs, particularly butyrate, and an increase in opportunistic pathogens, such as LPS-releasing bacteria [120,121]. Individuals with low gut bacterial diversity are more prone to conditions like overweight, obesity, dyslipidemia, IR, and low-grade inflammation [122]. Research also indicates that specific dietary interventions can modify the gut microbiome composition in obese individuals [123] (Figure 2). Consequent structural changes in the intestinal epithelium allow LPS and gut bacteria to enter the bloodstream in humans and mice [123,124,125,126,127]. This leads to an increase in plasma levels of gut bacteria and LPS, a condition known as metabolic endotoxemia [125]. LPS activates Toll-like receptor 4 (TLR4), subsequently promoting the activation of nuclear factor kappa B (NF-κB)-dependent transcription programs for various pro-inflammatory cytokines, such as IL-1β, IL-18, IL-6, IL-33, TNF-α, and interferon-gamma (IFN-γ). As a consequence, metabolic endotoxemia can contribute to chronic inflammation typically associated with obesity and related diseases [125]. Of note, previous findings have reported the presence of bacterial DNA and living bacteria in obese human AT, suggesting the host–microbial interaction as a possible link to the dysmetabolic and inflammatory status of obese patients [128].

Recently, a study conducted by Shantaram et al. provided multiple lines of evidence in support of the translocation of bacteria from the gastrointestinal tract of obese individuals as a contributor to neutrophilia in visceral AT (VAT) [12]. Obese subjects showed elevated plasma levels of LPS-binding protein and zonulin, and the count of neutrophils in VAT correlated with circulating LPS levels. Ampliseq analyses showed that LPS-responsive genes, such as *LITAF* (LPS-induced TNF factor), TLR2, TLR4, and CD14—proteins that bind to bacterial components [129,130,131]—were upregulated in VAT neutrophils compared to circulating neutrophils. In addition, the bacteria found in human VAT were found to primarily originate from the gut. Adipocytes from individuals with obesity expressed IL-8 at levels ten times higher than those in lean individuals, and LPS stimulated IL-8 gene expression in cultured adipocytes by nearly 150-fold. Therefore, IL-8 produced by adipocytes in response to bacteria likely played a significant role in attracting neutrophils to VAT.

SCFAs, including acetate, butyrate, and propionate, are believed to mediate the relationship between gut microbiota and immune responses [132]. These metabolites influence the epigenetic regulation of immune cells, including neutrophils. For example, SCFAs regulate the recruitment of neutrophils by modulating the synthesis of inflammatory substances such as IL-17 and TNF-α [118]. They stimulate the neutrophil G-protein-coupled receptor 43 (GPR43), which plays a key role in neutrophil chemotaxis [133]. Additionally, SCFAs can alter neutrophil functions, such as their ability to perform phagocytosis and generate ROS and nitric oxide (NO) [134,135]. While ROS are vital for eliminating bacteria and other pathogens, excessive production can damage body tissues. SCFAs help regulate ROS generation in neutrophils, supporting a healthy immune response [136]. However, the exact impact of SCFAs on neutrophil reactions to inflammatory substances like LPS is still not fully understood. Research has shown that SCFAs can significantly reduce TNF-α production in human neutrophils exposed to LPS [135], but they do not affect IL-8 production, indicating that LPS triggers TNF-α and IL-8 production through different pathways. A study by Li et al. [137] evaluated the role of butyrate in modulating NETs in an inflammatory bowel disease (IBD) mouse model. They found that butyrate improved mucosal inflammation by inhibiting neutrophil-associated immune responses, including NET formation. Butyrate has also been shown to inhibit in vitro neutrophil migration and the development of NETs in cells from IBD patients [137,138].

In summary, one fascinating way by which the gut microbiome influences the immune response is by regulating NET formation through SCFAs, which function effectively under specific concentrations and environmental conditions. SCFAs produced by the gut microbiome enter the bloodstream and affect AT physiology through mechanisms that may involve GPRs, also known as free fatty acid receptors (FFARs) [139]. FFAR2 (GPR43) selectively binds to acetate, while propionate and butyrate show high affinity for FFAR3 [140], thereby activating metabolic pathways associated with obesity and related diseases [141,142]. SCFAs also impact the intestinal environment by lowering colonic pH [143] and modulating microbial and colonic energy homeostasis [11,144]. They can counteract obesity-related inflammation [145] and regulate glucose and lipid metabolism [146,147]. Numerous studies suggest that SCFAs are promising therapeutic agents for managing obesity and metabolic disorders, especially due to their ability to modulate NET formation. However, further research is needed to fully understand the precise mechanisms through which SCFAs regulate this process.

Probiotic supplementation is recognized as an effective strategy for modulating gut microbiota [148,149,150] and increasing SCFA production in individuals with severe obesity [151]. By reshaping gut microbiota, reducing inflammation, and improving overall intestinal health, this approach can lower metabolic endotoxemia, thus helping to restore the function of peripheral tissues, such as AT [152,153]. Therefore, probiotics are valuable tools in the treatment of obesity and related diseases.

## 9. Concluding Remarks

Neutrophils are increasingly recognized as important early participants in the development of obesity and related diseases. In this review, we have compiled key studies that emphasize the role of neutrophils and their activation in the onset and progression of obesity and obesity-related conditions. We highlight their significant contribution to the inflammatory response triggered by HFDs. A better understanding of the causal relationship between neutrophils and obesity may lead to new management strategies for obesity-related inflammation, positioning neutrophils as a potential therapeutic target.

Emerging evidence indicates that NETosis is implicated in obesity and related diseases, such as T2D and NASH [94,98]. Excessive activation and inappropriate recruitment of NETs due to increased AT may contribute to alterations in the inflammatory profile associated with obesity. This can lead to both local and systemic inflammation, ultimately resulting in severe organ damage. However, the interaction between NETosis and chronic metabolic diseases is not fully understood, and there is a significant gap in comprehensive mechanistic studies.

Future research should aim to clarify the mechanisms of NETosis in various diseases and explore ways to regulate its generation and interruption. It is essential to remember that neutrophil phagocytosis is crucial for host defense, and any drugs targeting NET formation must not compromise the physiological functions of these cells. Clinical trials should be designed to evaluate the efficacy of NET-interfering drugs in obese patients. In conclusion, while the discovery of NETosis has significantly advanced our understanding of the pathophysiology and natural history of several diseases, further extensive research is necessary. Further studies should investigate the impact of NETs on different diseases and foster the development of targeted therapeutic strategies that minimally affect the host’s immune system.

## Figures and Tables

**Figure 1 ijms-25-13633-f001:**
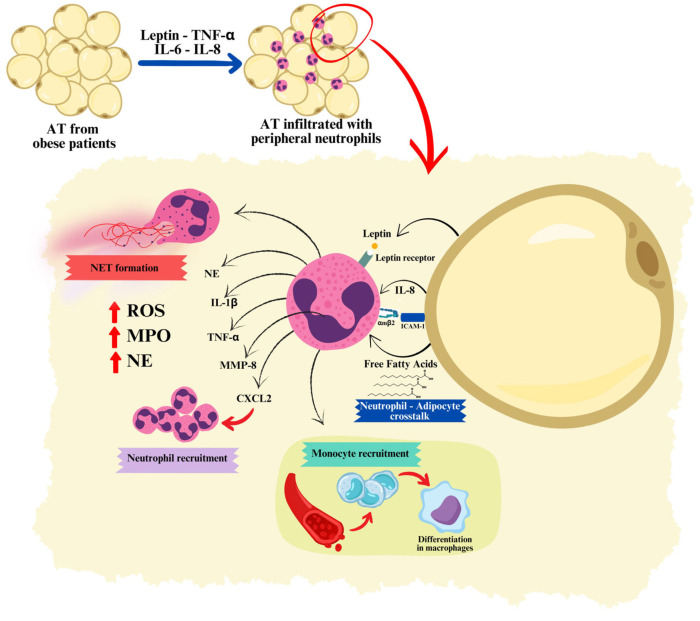
The emerging role of neutrophils as critical early players in the pathogenesis of obesity. Neutrophil recruitment by obese adipose tissue (AT), and bidirectional adipocyte–neutrophil crosstalk have a substantial impact on the inflammatory profile of AT through the production of a plethora of pro-inflammatory factors and NET formation. These images were created with https://www.canva.com/ (accessed on 12 November 2024).

**Figure 2 ijms-25-13633-f002:**
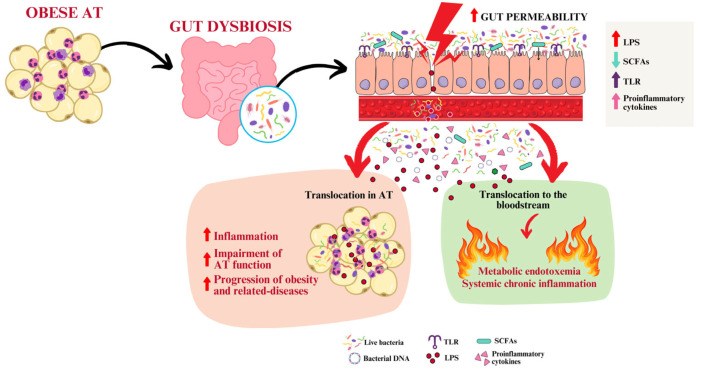
Gut dysbiosis and disruption of intestinal barrier permeability are linked to obesity. As a result, the translocation of live bacteria, bacterial DNA, LPS, and pro-inflammatory cytokines into peripheral circulation and adipose tissue worsens the inflammatory response. This exacerbates the progression of obesity and related diseases. The arrow symbols (pointing up and down) indicate the increase or decrease of molecules, specifically SCFAs, TLR, and proinflammatory cytokines. The colors of the arrows correspond to the colors of the symbols used. These images were created with https://www.canva.com/ (accessed on 12 November 2024).

**Table 1 ijms-25-13633-t001:** Main preclinical and clinical studies investigating the role of activated neutrophils in obesity and related diseases.

Preclinical and Clinical Studies	Role of Neutrophils	Reference
Animal models	HFD-fed mouse model. Early increase in the expression of AT MPO and neutrophil infiltration in response to an HFD. Once in the AT, neutrophils interact with adipocytes through the binding of integrin αMβ2 (Mac-1) on the neutrophils to ICAM-1 on the adipocytes. This interaction activates the neutrophils and prompts them to produce IL-1β and TNF-α, which further stimulate inflammation in the AT.	[9]
	HFD-fed mouse model. Sustained increased AT and liver neutrophil content. High levels of NE secretion and association to IR.	[48]
	HFD-fed mouse model. Neutrophil-derived cathelicidin (CRAMP) may enhance the adhesion and extravasation of monocytes into atherosclerotic arteries through direct or indirect chemotactic activities, making them important facilitators of monocyte/macrophage accumulation during the early stages of atherosclerosis.	[50]
	HFD-fed mouse model. Neutrophil activation and NE deposition into AT are correlated with significant higher levels of the chemoattractants MCP-1, IL-8, and markers of activated M1 macrophages such as F4/80, CD68, TNFα, and CD11c.	[15]
	HFD-fed mouse model. Neutrophil activation and neutrophil-derived NE in AT can promote the activation and infiltration of macrophages and increased inflammatory cytokine mRNA levels.	[51]
	p38γ/δ^−^/^−^ mice model. Mice lacking p38γ/δ in myeloid cells are protected against diet-induced fatty liver. This effect is due to defective migration of p38γ/δ-deficient neutrophils to the damaged liver, where they normally induce inflammation and metabolic changes. p38γ/δ KO and myeloid-specific p38γ/δ cKO mice are resistant to hepatic steatosis induced by HFD or methionine-choline-deficient diet.	[52]
	HFD-fed mouse model. Neutrophils and IL-1b are required for the efficient expression of chemotactic molecules in adipose tissue, which contributed to macrophage infiltration.	[38]
	HFD-fed rat model. Neutrophil-derived MPO affects microvascular IR or muscle metabolic insulin sensitivity.	[54]
	HFD-induced NASH mouse model; Caspase-1 knockout mice, which also miss caspase-11, and NE/PR3 knockout mice, were developed and intercrossed to obtain quadruple knockout mice (Casp1/Casp11/NE/PR3). Age-matched WT C57BL/6 mice were used as controls. Mice deficient in caspase-1, NE, and PR3 are protected from developing diet-induced weigh gain, liver steatosis, and AT inflammation when compared with controls.	[57]
Clinical studies	N. 37 severely obese subjects, N. 9 control subjects. Both circulating MPO and calprotectin levels are significantly increased in severely obese subjects as compared to healthy controls. Neutrophil-specific marker CD66b is significantly increased in severely obese individuals. Weight loss after bariatric surgery is associated with decreased plasma levels of inflammatory mediators.	[28]
	N. 40 Human liver specimens obtained during bariatric surgery from severely obese patients. Hepatic expression of CXC chemokines is associated with MPO-mediated oxidative damage along with increased expression of IL-8 and neutrophil sequestration in NASH.	[49]
	N. 223 obese children, N. 223 normal-weight children. MPO may serve as an early biomarker of inflammation linked to cardiovascular risk in obese prepubertal children, similar to CRP, MMP-9, and resistin.	[29]
	N. 14 morbidly obese women, N. 9 control lean women. Circulating levels of CXCL2 are approximately 3-fold higher in obese subjects than in controls. Analogously, CXCL2, IL-2, IL-6, IL-8, and other chemokines are significantly increased in subcutaneous AT from obese subjects compared with controls. Blood neutrophils from obese subjects release high levels of proinflammatory mediators, MPO, and ROS, promoting the initiation and perpetuation of the senescence of AT-ECs.	[35]
	N. 49 obese subjects, N. 46 lean subjects. Male obesity was associated with increased percentage of peripheral neutrophils and increased expression of neutrophil activation-related genes, including MPO and NE.	[30]
	N. 9 obese patients with NASH, N. 11 healthy control subjects. Expression of p38δ and p38γ is elevated in the liver from patients with NASH. p38γ/δ control neutrophil migration to the damaged liver. Migration of neutrophils to the liver is necessary for the development of steatosis.	[52]
	N. 112 T2D patients (N. 27 good glycemic control without complications (GC), N. 32 good glycemic control with complications (GCC), N. 21 poor glycemic control without complications (PC) and N. 32 poor glycemic control with complications (PCC), N. 34 non-diabetic volunteers (NGT). In neutrophils, the highest levels of the proinflammatory cytokines, TNFα, IL-6 and IFN-β mRNA are present in the GC subjects when compared to NGT control. RANTES mRNA expression is elevated in neutrophils obtained from the GC subjects. Also, expression of most TLR mRNAs is increased in neutrophils from GC subjects when compared to NGT. IL-6 and IFN-β mRNA levels are elevated in neutrophils from GC and PC and TNFα mRNA levels are elevated in neutrophils from GC group, compared with NGT.	[53]
	N. 25 obese subjects, N. 27 overweight subjects, N. 22 lean subjects. NE and MPO mRNA expressions in the peripheral blood leukocytes are upregulated in overweight and obese subjects as compared to lean subjects. No difference was found between overweight and obese groups. The NE and MPO mRNA levels show significant positive correlation with markers of cardiovascular disease including BMI, serum triglycerides, and atherogenic index of plasma.	[31]
	N. 18 obese subjects, N. 18 control healthy subjects. Obese patients show increased activity of peripheral blood neutrophils which exhibit higher ROS generation and release of cytokines, thus enhancing the extent of local and systemic inflammation. Following weight loss from gastric band surgery, there is a notable reduction in the levels of MPO and ENA-78, a neutrophil-activating peptide derived from epithelial cells that is elevated in inflamed tissues. The weight loss is also associated with decreased ROS production and lower cytokine release compared to healthy controls.	[58]
	N. 271 obese individuals with NASH, N. 41 patients with NASH, N. 401 obese with T2D patients and N. 205 lean healthy controls. PR3 and NE plasma concentrations are elevated both in individuals with liver steatosis and in patients with T2D when compared to lean healthy individuals and obese individuals and are associated with hsCRP, the marker of systemic inflammation. PR3 and NE concentrations in the liver tend to be higher in patients with advanced stages of NASH when compared to patients with mild disease.	[55]
	N. 162 obese children and adolescents, N. 73 overweight children and adolescents, N. 47 normal weight children and adolescents. Absolute neutrophil count is significantly higher in children with obesity (*p* = 0.002) compared to non-obese participants. A positive correlation between neutrophil count and waist circumference, fasting insulin, IR, and triglycerides. Neutrophils are not associated with glucose levels (fasting, during glucose tolerance test or as HbA1c) or the rest of the lipid profile. CRP trended with ANC overall. In addition, absolute neutrophil count, and CRP are associated with metabolic impairment and cardiovascular risk factors	[27]
	N. 30 NASH patients, N. 10 healthy control subjects. Peripheral blood neutrophils from NASH patients show, at baseline, high levels of ROS production as compared to controls. Neutrophils of NASH patients show a more active phenotype and strongly suppress CD4+ and CD8+ T cell proliferation and activation, suggesting that, in the presence of steatohepatitis, an immunological tolerance might take place, thus contributing to the progression of liver disease.	[56]
	N. 82 obese subjects, N. 14 control lean subjects. VAT neutrophils are more abundant in obese subjects compared to lean subjects and express more inflammation and activation-related genes compared to peripheral blood neutrophils. VAT neutrophils correlate with both circulating levels of LPS and insulin resistance, suggesting VAT neutrophil abundance is related to the presence of bacteria and influences systemic metabolism.	[12]

**Table 2 ijms-25-13633-t002:** Main studies investigating NETs in obesity and related diseases.

Preclinical and Clinical Studies	Role of NETs	References
Obese murine models	Increased NET accumulation in mesenteric arterial walls of obese mouse models may affect endothelial function.	[16]
	Increased systemic NET release is linked to obesity progression and an increased risk of venous thrombosis.	[87]
	NETs in the pancreas may induce an inflammatory response in ductal cells and promote the development of pro-tumorigenic cells both in vivo and vitro.	[107]
Obese patients and obese murine model	Elevated NET serum levels correlate with inflammatory markers such as IL-8, HSP90, and HSPE1.	[17]
Obese patients	Increased levels of plasmatic NETs in obese patients are correlated with BMI, waist, hips, glyco-metabolic markers and blood pressure.	[88]
Murine NASH model	Accumulation of NETs in liver tissue are identified as an early event in the development of NASH.	[89,90]
NASH patients and murine NASH model	NETs promote hepatic inflammation and contribute to the development and progression of hepatocellular carcinoma in NASH.	[91]
NASH patients	NET accumulation exerts cytotoxic effects on the ECs, converting them to a procoagulant and pro-inflammatory phenotype in NASH.	[94]
T2D patients and murine T2D models	Elevated levels of circulating NET markers are predictive of delayed healing in diabetic patients. In a diabetic mouse model, higher levels of NET markers and increased PAD4 activity are associated with wound nonhealing.	[103]
	The presence of NET components worsens ulcers in T2D patients. In a diabetic mouse model, increased skin PAD4 activity and NET markers hinder wound healing.	[104]
	Excessive deposition of NETs in the renal glomeruli of diabetic patients and mouse models cause injury to glomerular endothelial cells, contributing to the progression of diabetic kidney disease.	[106]
T2D patients	Hyperglycemia is associated with increased systemic levels of NET markers in T2D patients. In vitro results confirm that high glucose induces an increased release of NETs by white blood cells isolated from healthy donors.	[98]
	Findings support that NETs represent biomarkers in T2D. Increased NETs do not seem to result from poor glycemic control; rather, they are attributed to proinflammatory cytokines.	[96]
	Higher levels of systemic NET biomarkers are associated with T2D.	[97]

**Table 3 ijms-25-13633-t003:** Inhibition of NETs as a therapeutic approach in obesity and related diseases.

Active Agent/Drug with Anti-NET Potential	Mechanism of Action and Effect	Reference
Cl-amidine	NET inhibition; Promotion of wound healing in murine T2D model.	[104]
Cl-amidine and DNase I	NET inhibition; Reduction of the endothelial dysfunction through restoring vasodilation of the mesenteric arteries in obese mice.	[16]
DNase I	NET inhibition; Protection from HFD-induced liver injury and reduction of hepatic damage progression in NASH murine model.	[89,90]
	NET inhibition; Reduction of liver inflammatory profile and development/progression of hepatocellular carcinoma in NASH murine model.	[91]
	NET inhibition; Reverting the damaging actions of NETs in the procoagulant and proinflammatory phenotype in NASH patients.	[94]
	NET inhibition; Promotion of wound healing and reduction of NET-driven chronic inflammation in murine T2D models.	[103]
	NET inhibition; Reduction of glomerular EC injury and improvement of diabetic kidney disease in murine model.	[106]
DNase I and Metformin	NET inhibition; Combined treatment with Metformin and DNase I significantly reverses the pro-tumorigenic effects of NETs.	[107]
Metformin	NET inhibition; Metformin treatment reduces the levels of NET formation independently of glucose control in T2D patients.	[96,99]
	NET inhibition; Metformin intake before surgery significantly improves the colorectal cancer-associated outcomes of diabetic patients to a level equal to or even better than that of non-diabetic patients.	[95]
Silybin	NET inhibition; Reduction of HFD-induced liver injury in NASH murine model.	[90]

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
