# Peer review of "The Evolving Role of Neutrophils and Neutrophil Extracellular Traps (NETs) in Obesity and Related Diseases: Recent Insights and Advances"

_ijms, 2024, doi:10.3390/ijms252413633_

Round 1
Reviewer 1 Report
Comments and Suggestions for Authors
The authors summarize recent progress in researched regarding NET formation and its involvement in obesity and related metabolic diseases in this review paper. The authors followed a great number of references that are published within the last 20 years. NET formation is well known to related to rheumatoid arthritis or thrombosis, but it has paid attention as possible cause of many other diseases. Thus, the topic of this review is of interesting and timely. However, this reviewer would like to point out that this manuscript needs to focus on this main topic, and there are a few chapters that are slightly out of focus. I would like suggest several concerns that should be improved.
Specific points.
1. Line 111-121. The authors referred to low-density neutrophils (LDN)in this paragraph. However, I think this not directly related to NET formation. Actually, LDN is not appeared in the other parts of this manuscript. I think this paragraph can be omitted.
2. Chapter 4. The authors described that neutrophil-to lymphocyte ration (NLR) is a goof maker for inflammatory responses. Yes, this is true, but again, I could not see the relationship between NLR and NET formation in this chapter. Also, the authors did not show subjective information (critical number s of NLR, statistical significance, etc.), Impact of this chapter to the main topic, NET formation, seems to be low.
3. Chapter 5. The authors described that NET formation could be involved in obesity-induced inflammation. In many tissues inflammation responses occur by the action of many types of leukocytes including macrophages. In adipose tissues, it is well-known that macrophages filtrated into adipose tissues act as inflammatory stimulator. I think the roles of macrophages as well as neutrophils in adipose tissue should be discussed.
4. Line 272-275. I wonder whether increase in MCP-1, strong chemoattractant for monocyte/macrophages could support involvement of neutrophils and NET formation,
5. Line 335-343. Please indicate direct evidence (refer to a study) showing metformin inhibits NET formation. If the Inhibition of NETs is caused by blood sugar levels, it should be fare to discuss about the effect of other anti-DM drugs and why metformin is positive.
6. Chapter 10. The Concluding remarks is too long, and it lacks focus. I do not think it is good to mention many other diseases in this section. I think it is better to shorten this chapter.
7. Overall, the authors followed many references, which is good. However, description of each study should be more subjective. In some cased utilizing actual numbers such as fold-changes, statistical differences, concentrations, etc. would be very helpful for readers to figure out the reactions and estimate the credibility of a possibility.
Author Response
Comments and Suggestions for Authors
The authors summarize recent progress in researched regarding NET formation and its involvement in obesity and related metabolic diseases in this review paper. The authors followed a great number of references that are published within the last 20 years. NET formation is well known to related to rheumatoid arthritis or thrombosis, but it has paid attention as possible cause of many other diseases. Thus, the topic of this review is of interesting and timely. However, this reviewer would like to point out that this manuscript needs to focus on this main topic, and there are a few chapters that are slightly out of focus. I would like suggest several concerns that should be improved.
Authors’ reply: The authors would like to express their sincere gratitude to Rev#1 for thoroughly evaluating the manuscript and providing valuable comments and criticisms.
We have carefully revised the manuscript in response to these insights.
Below, we present a point-by-point list of the changes made, addressing each specific issue and suggestion from Rev#1.
Specific points.
- Line 111-121. The authors referred to low-density neutrophils (LDN) in this paragraph. However, I think this not directly related to NET formation. Actually, LDN is not appeared in the other parts of this manuscript. I think this paragraph can be omitted.
Authors’ reply: As suggested by Rev#1, the marked paragraph has been omitted.
- Chapter 4. The authors described that neutrophil-to lymphocyte ration (NLR) is a goof maker for inflammatory responses. Yes, this is true, but again, I could not see the relationship between NLR and NET formation in this chapter. Also, the authors did not show subjective information (critical number s of NLR, statistical significance, etc.), Impact of this chapter to the main topic, NET formation, seems to be low.
Authors’ reply: As suggested by Rev#1, Chapter 4 has been omitted.
- Chapter 5. The authors described that NET formation could be involved in obesity-induced inflammation. In many tissues inflammation responses occur by the action of many types of leukocytes including macrophages. In adipose tissues, it is well-known that macrophages filtrated into adipose tissues act as inflammatory stimulator. I think the roles of macrophages as well as neutrophils in adipose tissue should be discussed.
Authors’ reply: We thank Rev#1 for the valuable suggestion. In response, the authors have added a section to Chapter 5 that discusses the crucial role of macrophages in obesity. This section particularly emphasizes the impact of neutrophils on the recruitment, retention, and proliferation of macrophages in obese adipose tissue.
- Line 272-275. I wonder whether increase in MCP-1, strong chemoattractant for monocyte/macrophages could support involvement of neutrophils and NET formation
Authors’ reply: In response to the question posed by Rev#1 and referencing the cited article, we have added the following sentence (Line 275): The authors suggest that since MCP-1 expression may be connected to cardiovascular disease progression associated with obesity, their findings provide further support for the hypothesis that NET formation is involved in the inflammatory processes linked to DIO.
- Line 335-343. Please indicate direct evidence (refer to a study) showing metformin inhibits NET formation. If the Inhibition of NETs is caused by blood sugar levels, it should be fare to discuss about the effect of other anti-DM drugs and why metformin is positive.
Authors’ reply: We sincerely thank Rev#1 for the valuable suggestion and the stimulating question posed, which has allowed us to enhance Chapter 8 with additional information about metformin. Specifically, we have incorporated the latest evidence indicating that metformin has been shown to significantly reduce the formation of NETs associated with cardiovascular disorders related to diabetes mellitus, independent of its role as an anti-hyperglycemic agent.
We also included the description of a recent specific study showing the ability of metformin to inhibit the NET formation by human neutrophils stimulated with either PMA or HMGB1 (10.3390/ijms23169134).
- Chapter 10. The Concluding remarks is too long, and it lacks focus. I do not think it is good to mention many other diseases in this section. I think it is better to shorten this chapter.
Authors’ reply: We thank Rev#1 for the valuable suggestion. As recommended, the authors have significantly condensed the Concluding Remarks section, removing references to various other diseases.
- Overall, the authors followed many references, which is good. However, description of each study should be more subjective. In some cased utilizing actual numbers such as fold-changes, statistical differences, concentrations, etc. would be very helpful for readers to figure out the reactions and estimate the credibility of a possibility.
Authors’ reply: We sincerely thank Rev#1 for the valuable suggestion. In response to this recommendation, we have revised the manuscript to include sentences that emphasize the specificity of the results obtained from the cited studies, particularly those listed in what are now Tables 2 and 3. In the original MS, these tables were referred to as Tables 1 and 2.
Reviewer 2 Report
Comments and Suggestions for Authors
In the present review article by Altamura et al., the authors discuss the recent findings and ongoing research on the emerging role of neutrophils and one of the crucial functions of neutrophils: extracellular trap formation. The topic covered is relevant and of interest to the current research focus. These short-lived cells have a larger impact on the immune system and in the context of obesity which is one of the underlying causes of various diseases, this topic becomes more relevant. I like the overall presentation and flow of the review. However, I have some suggestions for the author’s consideration:
1) While reading the title of the review it seems that this review involves a discussion of diverse functions and roles of neutrophils with a special focus on NET formation as its function, but the concluding remarks only highlight the importance and implication of NET formation in obesity and related disease.
2) Since the review also encompasses other functions of neutrophils like chemotaxis, infiltration, ROS generation, and degranulation. It would be relevant if authors could create a table summarizing the recent findings/studies investigating these functions of neutrophils in obesity and related diseases.
Author Response
Comments and Suggestions for Authors
In the present review article by Altamura et al., the authors discuss the recent findings and ongoing research on the emerging role of neutrophils and one of the crucial functions of neutrophils: extracellular trap formation. The topic covered is relevant and of interest to the current research focus. These short-lived cells have a larger impact on the immune system and in the context of obesity which is one of the underlying causes of various diseases, this topic becomes more relevant. I like the overall presentation and flow of the review.
Authors’ reply: The authors would like to express their sincere gratitude to Rev#2 for thoroughly evaluating the manuscript and for providing valuable comments and criticisms.
We have carefully revised the manuscript in response to these insights.
Below, we present a point-by-point list of the changes made, addressing each specific issue and suggestion from the Rev#2
However, I have some suggestions for the author’s consideration:
1) While reading the title of the review it seems that this review involves a discussion of diverse functions and roles of neutrophils with a special focus on NET formation as its function, but the concluding remarks only highlight the importance and implication of NET formation in obesity and related disease.
Authors’ reply: The authors thank Rev#2 for the valuable suggestion. The "Concluding Remarks" section has been revised to include a paragraph summarizing the importance and implications of neutrophils in obesity and related diseases.
2) Since the review also encompasses other functions of neutrophils like chemotaxis, infiltration, ROS generation, and degranulation. It would be relevant if authors could create a table summarizing the recent findings/studies investigating these functions of neutrophils in obesity and related diseases.
Authors’ reply: We would like to express our gratitude to Reviewer #2 also for this valuable suggestion, which has allowed us to significantly improve the revised version of the manuscript. In response to this feedback, we have added a new Table 1 titled "Main preclinical and clinical studies investigating the role of activated neutrophils in obesity and related diseases." Consequently, the original Tables 1 and 2 in the manuscript have been renamed Tables 2 and 3, respectively.
Round 2
Reviewer 1 Report
Comments and Suggestions for Authors
The manuscript was extensively revised, and it was greatly improved. I think it is now suitable for publication in IJMS.